# The Language of Pain in the Hypermobile Ehlers–Danlos Syndrome: Metaphors as a Key to Understanding the Experience of Pain and as a Rehabilitation Tool

**DOI:** 10.3390/brainsci13071042

**Published:** 2023-07-08

**Authors:** Filippo Camerota, Rachele Mariani, Giulia Cordiano, Michela Di Trani, Valentina Lodato, Alessandro Ferraris, Massimo Pasquini, Claudia Celletti

**Affiliations:** 1Physical Medicine and Rehabilitation, Umberto I University Hospital, 00161 Rome, Italy; clacelletti@gmail.com; 2Department of Dynamic and Clinical Psychology, and Health Studies Sapienza, University of Rome, 00133 Rome, Italy; rachele.mariani@uniroma1.it (R.M.); michela.ditrani@uniroma1.it (M.D.T.); 3Fondazione Don Carlo Gnocchi, 00186 Rome, Italy; giuliacordiano@gmail.com; 4Laboratory of Medical Genetics, Department of Experimental Medicine, San Camillo Forlanini Hospital, Sapienza University of Rome, 00185 Rome, Italy; vlodato@scamilloforlanini.rm.it (V.L.); aferraris@scamilloforlanini.rm.it (A.F.); 5Department of Human Neurosciences, Sapienza University of Rome, 00185 Rome, Italy; massimo.pasquini@uniroma1.it

**Keywords:** chronic pain, hypermobile Ehlers–Danlos syndrome, linguistic content analysis, metaphors, pain, symbolizing process

## Abstract

Ehlers–Danlos syndromes are a heterogeneous group of Heritable Connective Tissue Disorders characterized by joint hypermobility, skin hyperextensibility, and tissue fragility. Among the different types, the hypermobile Ehlers–Danlos syndrome is the most frequent and includes generalized joint hypermobility as the major diagnostic criterion. Joint hypermobility in hypermobile Ehlers–Danlos syndrome is often associated with pain that does not always allow the use of effective pain-reducing treatments. Patients with hEDS constantly describe their pain in detail. Eighty-nine patients with hEDS diagnoses were recruited and evaluated. They were asked to describe their pain in writing. The texts were examined through Linguistic Inquiry and Word Count. Correlational analyses were conducted between pain perception and language. A comparison of high/low pain perception and the quality of metaphors was carried out. The results showed that language quality varies depending on how much pain is perceived. The greater the pain is perceived, the lesser the positive effects and the greater the negative effects and dehumanizing metaphors are being used. Moreover, a greater pain seems to be related to a verbal experience of greater isolation and less self-care. In conclusion, the use of metaphors is a useful tool for examining illness experience and may help clinicians in the rehabilitation program.

## 1. Introduction 

Ehlers–Danlos syndromes (EDSs) are a heterogeneous group of rare hereditary disorders of connective tissue (HCTDs) with common features including joint hypermobility, skin hyperextensibility, and tissue fragility [1]. The most recent clinical classification, introduced in 2017, and subsequent updates now recognize 14 different subtypes of EDS [1,2,3,4]. The hypermobile type (hEDS) is generally considered to be the most frequent in the general population, followed by the classical (cEDS) and the vascular (vEDS) types, while others appear mostly ultrararely. A detailed revision of the clinical diagnostic criteria for hEDS was also introduced in 2017 to allow a better distinction from other EDS types and other syndromic disorders that can also include generalized joint hypermobility as a major clinical feature [2]. In the meanwhile, the term “Hypermobility Spectrum Disorders (HSD)” was proposed to include clinically relevant joint hypermobility, mostly symptomatic for secondary manifestations or complications, in patients that did not fulfil the clinical diagnostic criteria either for EDS, other HCTDs, or other recognizable syndromes [5]. The diagnosis of hEDS still relies exclusively on a combination of clinical inclusion criteria (generalized joint hypermobility, chronic pain and/or joint instability, signs of multisystemic connective tissue involvement, positive family history for hEDS) and some exclusion criteria. This happens because the genetic etiology of hEDS is still largely unknown and specific and sensible genetic tests are not currently available to support the clinical diagnosis [1]. Chronic pain is a fundamental criterion for diagnosis of hEDS, able to interfere with many aspects; in fact, concerning the main symptoms of joint hypermobility in hEDS, chronic localized and/or diffuse pain is most often referred from patients and is also considered to be the major determinant of the perceived reduced quality of life [6,7] and, in particular, influences the role of emotional and mental health elements.

Different papers have analyzed chronic pain in hEDS and HSD (in the literature before 2017, the diagnosis of hEDS or Joint Hypermobility Syndrome was obtained in many patients that are now expected to receive a diagnosis of hEDS or HSD, using the current criteria and nosology) trying to explain its pathogenesis in order to better address an effective treatment [8,9,10,11,12]. Pain in hEDS and HSD is considered multifactorial, partly related to hypermobility, joint instability, trauma, and previous surgery, and associated with moderate to severe impairment in daily functioning [8,13]. Pain can also arise from damage to the somatosensory system itself, and it should be a neuropathic pain [14]. When pain persists for more than 3–6 months or resurfaces after the tissue damage has resolved, neuroplastic changes in the pain processing pathways may lead to a hypersensitive state of the somatosensory system, a phenomenon called central sensitization [15,16].

From an observational point of view, patients with hEDS always talk about pain using a lot of descriptions and sometimes also with drawings [9]. Chronic pain is defined as pain that lasts for more than three months, and it is linked to many psychological comorbidities, including substance abuse, depression, and anxiety. In the absence of objective pain assessment tools, those who are experiencing it must rely on language and nonverbal pain behaviors like facial expressions to express their subjective experiences. Although pain is inherently elusive and private, people often better succeed in communicating their internal experiences with others using metaphors, which take on symbolic significance and reflect their internal state. This allows for external communicability and activating greater empathy in others [17]. A well-documented linguistic tool for communicating grief is metaphor [18,19]. This provides access to a more shared construction of meaning. The analysis of metaphors facilitates the exploration of how each person makes sense of the world [20].

In order to encourage the elaboration of traumatic experiences or chronic disease, the Expressive Writing Intervention (EWI), a technique that focuses on written emotional expression, initially required participants to write about pasted-in experiences [21,22]. In later research, the writing prompts were broadened to include additional particular challenging events that had positive effects on other healthy and clinical groups [23,24,25,26]. To date, there are no standard instructions for Expressive Writing Intervention, but the topic should be formulated depending on the specific subjects to whom it is addressed, even with the possibility to suggest instructions focused on a present difficult experience, dedicated to positive expectations for the future, and proposing a different writing topic for each session [23]. In the present study, this writing technique was used to stimulate the production of symbolic thinking in relation to the perception of chronic pain triggered by hEDS. It was then proposed to describe the pain experienced from illness through written metaphors. Several studies have made explicit the richness with which the metaphorical stimulus can induce a reorganization of affective experience and a resignification of the meaning of life [27,28,29].

This process promotes a different representation of the events in the memory, the memory is simplified, and it can be easily recalled in the mind, and these cognitive changes imply a different understanding of the experience and a change of perspective on the event [30]. To analyze metaphors, the Linguistic Inquiry and Word Count (LIWC) [31], a computerized text analysis program, has been applied. Linguistic analysis by LIWC was initially specifically created to analyze essays from Expressive Writing Intervention (EWI) studies, and it is also routinely used in analyzing psychologically meaningful writings [32,33]. The Linguistic Inquiry and Word Count program was also used to explore the association between word usage with various health and behavioral outcomes in expressive writing subjects [21]. In a recent study, the LIWC was applied to written descriptions of metaphors of life experiences, finding that this process can help to reformulate one’s own experiences [34,35]. In conclusion, computerized linguistic content analysis represents a possibility to deeply understand its role in the writing process.

The purpose of the present work was to test how the constant perception of pain interferes with symbolic and representational processes.

Our hypothesis is that chronic pain that exceeds tolerability produces narrative productions that are clinically different from more tolerable pain.

We decided to analyze the use of metaphors for describing pain in patients affected by hEDS, asking them to write down how they relate to their perception of their pain by responding to the following stimulus, “my pain is.....” Standardized clinical scale has been associated.

## 2. Materials and Methods

### 2.1. Participants

Participants were recruited over a three-month period (January 2021–March 2021) through the “Associazione per la Ricerca, Cura ed Assistenza Sindromi di Ehlers Danlos” (ARCASED Association), the “Clinici Ehlers-Danlos Italia” (CEDI Onlus Association), and the “Associazione Italiana per la Sindrome di Ehlers-Danlos” (AISED Association).

This cross-sectional study was conducted by sending, through the associations, an e-mail regarding the possibility to participate to this study to a list of 125 patients; among them, eighty-nine patients (79 females and 10 males, mean age 40 ± 13.11) with a diagnosis of hEDS accepted to participate and were recruited. Patients studied were chosen if they had chronic pain. Moreover, all participants were included if they (1) were 15 years of age or older, (2) had a diagnosis of hEDS made by a clinical geneticist following the 2017 diagnostic criteria and classification [2] at least one year earlier, and (3) spoke the Italian language. Participation was free, without any fee. This research was approved by the local institutional review board (REF.CE 4789).

### 2.2. Procedure

All the patients were invited to answer a survey and complete specific questionnaires.

The survey was custom-built and self-administered, taking about fifteen minutes to be completed. It was made clear to participants that it was anonymous and voluntary. The survey was hosted on Google Forms, which automatically collected the answers in order of arrival and transferred them onto a spreadsheet.

The final version of the questionnaire addressed the following areas: (a) demographic questions including age, identifying gender, and current employment status, (b) pain data collection through the McGill Pain Questionnaire, (c) metaphors used to define pain—in order to minimize the conceptual ambiguity, the general definition of metaphor was clarified and two examples were provided—and (d) psychopathological correlates through. the Zung Self-rating Anxiety Scale (SAS) and the Zung Self-rating Depression Scale (SDS). Then, the written metaphors were analyzed by computerized linguistic and content measures, such as Linguistic Inquiry and Word Count (LIWC).

### 2.3. Measures

Evaluation has been done using clinical scale and Linguistic Content Analysis.

### 2.4. Clinical Scale

*McGill Pain Questionnaire (MGPQ, Italian version)*: The McGill Pain Questionnaire (MPQ) measures the sensory, affective, and evaluative aspects of the pain experience. It consists of 78 pain descriptors, which are categorized into 20 groups, to evaluate the major dimensions of pain quality. Each of the 78 words is assigned a rank value. Only the total pain rating is considered. This value is obtained by adding the score assigned to each single pain characteristic considering the word used to represent the lowest pain intensity as 1, the next highest intensity as 2, and so on [36].

*Zung Self-rating Depression Scale (SDS):* Zung’s Self-rating Depression Scale (SDS) is widely used to assess depression [37]. It consists of 20 self-rated questions, with each item rated on a 4-point scale ranging from 1 (a little of the time) to 4 (most of the time). The total score is acquired by multiplying the raw score by 1.25. A higher total score indicates a more severe level of depression. An SDS score of 50 (raw score = 40) suggests clinically significant symptoms [38].

*Zung Self-rating Anxiety Scale (SAS)*: is a scale used to discriminate anxiety from mood disorders [39]. It is a 20-item Likert scale, in which items tap psychological and physiological symptoms; each item rated on a 4-point scale ranging from 1 (none, or a little of the time) to 4 (most, or all the time). The conversion of a total scale raw score (with a potential range of 20 to 80) to an index score with a potential range of 25 to 100 is derived by dividing the sum of the values (raw scores) obtained on the 20 items by the maximum possible score of 80, converted to a decimal and multiplied by 100. The total score of a value <50 indicates a no-anxiety condition, a minimal to mild anxiety is indicated by a score between 50 and 59 points, a moderate to marked anxiety is indicated between 60 and 69, and a condition of great anxiety is indicated if the score is more than 70.

### 2.5. Linguistic Content Analysis

#### Linguistic Inquiry and Word Count (Pennebaker, Booth, & Francis, 2015)

The Linguistic Inquiry and Word Count (LIWC) is a computerized program aimed to analyze data related to the language used in writing reports. The LIWC program includes the main text analysis module along with a group of built-in dictionaries. LIWC reads written or transcribed verbal texts, then compares each word in the text against a user-defined dictionary. After the processing module has read and accounted for all words in each text, it calculates the percentage of total words that match each of the dictionary categories. LIWC2015 v1.6 software has been used together with the Italian LIWC_2007 Dictionaries. Specific word categories were chosen for the purpose of the study: Social Processes Friends (pal, buddy, or coworker), Family (mom, brother, cousin), Humans (boy, woman, group); Affective Processes: Positive Emotions (happy, pretty, good), Negative Emotions (hate, worthless, enemy); Anxiety (nervous, afraid, tense); Anger (hate, kill, pissed) Sadness (grief, cry, sad); Past Time; Present Time; Future Time; Cognitive Processes: Inclusive (with, and, include); Exclusive (but, except, without); Personal concerns: Leisure; Home; Body Care.

## 3. Statistical Analyses

All statistical analyses were performed using the Statistical Package for Social Science, version 24 (SPSS version 24, Armonk, NY, USA). Data are reported as means and standard deviation for continuous variables and as percentages for discrete variables. To analyze metaphors in relation to pain experience, the group was divided using the cut-off of 35 points for the McGill Pain Questionnaire, a total index that is the mean value of pain in a population of patients with primary fibromyalgia [40], in order to differentiate into two groups: the first with high pain (MGPQ ≥ 36) and the second with low pain (MGPQ ≤ 35).

An independent T test was applied to compare linguistic measures. Pearson correlation was used to verify the relationship between linguistic measures and symptom measures. A *p* value < 0.05 was considered significant.

## 4. Results

Eighty-nine patients accepted to participate and were evaluated; the mean age was 40 (min. 15, max. 71) with a prevalence of females (79 F/10 M).

The mean values of the different scales used to evaluate pain, anxiety, and depression in these patients are summarized in Table 1.

Using the pain cut-off indicated, we divided the sample into two groups and analyzed the Linguistic Inquiry and Word Count (LIWC) language measures. The hypothesis developed was that chronic pain perception afflicts patients by conditioning the way they develop symbolic and linguistic processes. This hypothesis of ours can be said to be confirmed. The use of metaphors in the clinical group for pain perception turns out to be significantly different from the group with low pain perception for many linguistic parameters (Table 2).

Bivariate correlation analysis on the whole sample N. 89 between the linguistic and clinical measures of anxiety, depression, and pain scale specifically shows that the depression scale correlates negatively with positive emotions (r −0.252 *p* < 0.05) and body care (r −0.243 *p* < 0.05), and positively with anger (r 243 *p* < 0.05). The anxiety scale correlates negatively with friends (r −233 *p* < 0.05), humans (r −305 *p* < 0.05), and body care (r −256 *p* < 0.05). Finally, pain correlates negatively like anxiety, with the same language categories, friends (r −245 *p* < 0.05), humans (r −305 *p* < 0.02), and body-care (r −441 *p* < 0.01).

## 5. Discussion

All patients were asked to respond in writing to the following question, trying to make explicit a metaphor indicating “My pain is…” Correlation analysis shows that the clinical scales of anxiety, depression, and pain are related to specific scales of the human/nonhuman relationship. Depression is related to anger and lower positive feelings, elements consistent with the depressive dimension [41]. Anxiety and pain perception correlate with the same scales, highlighting how greater anxiety and greater pain tend to bring out a more alien and dehumanized narrative. By deepening this data by comparing the two groups, this element is confirmed. The groups with high and low pain perceptions used this stimulus in significantly different ways. The perception of lower pain is expressed by the metaphor of a presence in life as a friend/partner to live with. It is identified as a “companion”, a friend who never leaves you but is tolerated. This consistently repeated description is what leads to differentiating the low pain perception group with more words of human, friend, and positive emotions. “*A burning fire…A dull and annoying travel friend*” or “*My partner…sometimes silent sometimes not*” or “A troublesome companion in life: ‘*When it is most acute, I need to take care of him in my daily life…A troublesome master of life*’”. In other words, a more tolerable perception of pain leads it to be experienced as a friend who is loved and cared for. This element is confirmed both in the correlations and in the differences between groups.

Once pain exceeds the tolerability indices, however, the situation changes completely. Pain is metaphorized as a stranger, a dehumanizing element, preventing pleasure and activities. Finally, the *inclusive* category denotes the impossibility of excluding it from one’s life. Pain is, in fact, an element that connotes existence. In patients with clinical pain perception, they represent pain as an element to be kept “out”, to fight, an intrusive presence: “*A rodent eating viscera*”, “*A weight crushing me, a corset/armor constricting from head to toe*” or again “*Ice on my face*”. Thus, it seems that the level of chronicity of pain perception in the metaphorical dimension drives an inside/outside struggle; where pain crosses the tolerance level, it becomes an invader. The prevailing linguistic metaphor of the group with more pain is a dehumanized experience; pain, from a life companion, becomes the enemy from which to succeed in freeing oneself. An enemy so strong that it ruins one’s life and one’s pleasure. It could be concluded that the pain metaphors in this group of patients with a chronic illness represent how the relationship with pain is like a love affair, that is, it can be tolerated with flaws, but past a certain limit, it becomes the enemy to be divorced or to die from (Table 3). An additional important difference in the lexicon of the two investigated groups results from future perspective. The absence of conjugations toward the future seems to result in a squashing of the timeline, losing the perspective of something that may come to pass. Such a reduction amplifies the perception of immobility and painful eternity.

Pain is described with strong metaphors, and the more dehumanizing it feels, the stronger and more chronic the pain experience. Thus, a close relationship seems to emerge between pain perception and the quality of metaphoric content. This finding agrees with previous studies in which the use of symbolic language allows for embodied communication and a greater emotional vehicle for others [42,43,44]. Language is, in itself, an element that allows for a possible reframing of the experience of illness and the communicability of one’s pain.

## 6. Conclusions

To the best of our knowledge, this is the first survey that investigates the use of language and metaphors in a cohort of patients affected by hEDS.

Our study confirmed that patients with symptomatic (generalized) joint hypermobility with a clinical diagnosis of hEDS use significantly different linguistic metaphors in relation to the amount of perceived pain. The experience of pain experienced by hEDS patients is constant; the ability to give representable meaning to pain facilitates communicability with the outside world. The experience of pain is often experienced as a punishing, isolating element in a person’s silent suffering. Psychological understanding of others’ pain comes through the communicability of the experience. The attempt to make sense, of pain and emotional experience, is among the foundations of psychological care through the “symbolizing process”. The use of metaphors as a clinical and therapeutic tool for the symbolization and verbalization of pain is fundamental to accessing an experience of communicability and understanding of the other [25].

Language is, therefore, a therapeutic vehicle, that could be used for interventions aimed at integrating the mind–body relationship. Several studies analyzing linguistic processes, for example, have shown that being able to share, through written narratives or dream sharing, traumatic and painful experiences enabled people to better tolerate the painful experience [45,46].

In this study, patients with a higher level of pain tended to use emotional language with external references (“a knife”, “a bomb”) that might reflect a sense of helplessness and detachment from their bodies, as they had to resort to describing it as an aggression from the outside rather than an inner discomfort. These findings are consistent with previous research on pain metaphors [47,48], and it is likely that the personification of pain as an external force gives patients the opportunity to create an objectifiable enemy to fight, thus isolating the sick part from the healthy part within the body.

In contrast to Munday’s findings [47], none of our participants referred to everyday experiences of pain (such as illness or accidents), as if to argue that common events shared by listeners could not adequately describe the complexity of chronic pain.

Integrated interventions aimed at the expressiveness of the internal world and the symbolization of disease processes would allow for transformative thinking about the experience of pain. This study highlights how refining psychological interventions aimed at storytelling and expression of one’s pain experience would facilitate sharing and adherence to treatment [49,50]. The findings confirm that chronic pain has a complex and disorganized impact on people’s lives, making it difficult to organize a rehabilitative treatment.

In the rehabilitation field, the use of metaphors is often used but not always interpreted; if used, they are, instead, an important tool that may help the rehabilitator understand the type of pain and, especially, to organize the exercise program. Pain has a strong correlation with cognitive, sensitive, emotional elements, and memory; often, it is the only element for the patient, to recognize his own body, so exercise proposed to the patients needs to have cognitive elements [51], able to reduce fatigue and improve function [52].

The results achieved by this preliminary study provide broad support for the development of psychological interventions for chronic pain patient populations. The loneliness and sense of estrangement that pain produces, well-represented by metaphors, urge specific clinical and psychological care for patients. Emotional support and well-integrated clinical intervention would not only facilitate patient compliance but also enhance resources for coping with the pain experience. To this end, new interventions using symbolic and affective processes integrated with medical care are hypothesized to enhance their effectiveness.

## 7. Limitations

This study has several limitations that should be taken into consideration. The number of participants does not allow generalizations to be made explicit, and the absence of a control group does not allow the results to be said to be specific. In the future, it would be useful to explore the emotional and symbolic regulation processes related to pain by comparing different types of chronic pain.

## 8. Acknowledgments

Two of the authors of this publication (A.F. and V.L.) are members of the European Reference Network Connective Tissue and Musculoskeletal Diseases (ERN ReCONNET)—Project ID No 739543.

## Figures and Tables

**Table 1 brainsci-13-01042-t001:** The mean value of the McGill Pain Questionnaire, the Zung Anxiety scale, and the Zung depression scale.

MGPQ	Total Index (Mean ± s.d.)	N° Patients with High Pain (≥36)/%		N° Patients with Low Pain (≤35)/%	
	46.65 ± 13.55	71/79.8		18/20.2	
	Mean value	N° patients < 50 (absent)/total %	N° patients 50–59 (minimal to mild)/total %	N° patients 60–69 (moderate to marked)/total %	N° patients > 70 (great)/total %
Zung Anxiety score	64.23 ± 11.73	10/89 11.24%	19/89 21.34	30/89 33.71	30/89 33.71
Zung Depression score	61.03 ±12.09	18/89 20.21%	20/89 22.5%	26/89 29.2%	25/89 28.09%

MGPQ: McGill Pain Questionnaire.

**Table 2 brainsci-13-01042-t002:** Independent T test in linguistic measures of Linguistic Inquiry and Word Count (LIWC) indexes, between “Low-High pain perception”.

Variable	High Pain (n.71) M ds	Low Pain (n.18) M ds	T	P
LIWC				
Friends	0.28→1.41	2.17→4.51	−3.024	0.003 *
Family	0.02→0.17	0.00→0.00	0.501	0.617
Humans	0.24→1.01	2.42→4.78	−3.601	0.001 *
Positive Emotions	0.00→0.00	0.70→2.02	−2.945	0.004 *
Negative Emotions	3.90→6.64	2.63→5.16	0.758	0.451
Anxiety	0.16→1.04	0.23→0.98	−0.260	0.796
Anger	1.03→2.51	0.25→1.07	1.288	0.05
Sadness	1.89→4.16	0.00→0.00	−2.645	0.006 *
Past Time	0.56→2.58	0.15→0.66	0.660	0.511
Present Time	4.76→5.83	4.28→5.96	0.314	0.755
Future Time	0.00→0.00	0.21→0.87	−2.021	0.007 *
Inclusive	0.72→1.64	0.00→0.00	3.684	0.000 *
Exclusive	6.39→6.48	4.33→5.720	1.234	0.221
Leisure	0.26→1.33	1.40→4.29	−1.908	0.036 *
Home	0.13→1.08	1.39→4.28	−2.237	0.028 *
Work	0.210→0.70	0.89→2.07	−2.182	0.032 *
Body care	0.000→0.00	0.666→1.020	−1.974	0.033 *

Greed of freedom 87; * *p* < 0.05

**Table 3 brainsci-13-01042-t003:** Examples of metaphors for the low pain and high pain groups, “My pain is…”

Low Pain Group	High Pain Group
an annoying companion of life	a bomb bursting inside
mine, like a brand. suppressible with medicine, tolerated and listened to now that I know why. I see it change, listen to it move, like an ant	a knife that haunts
light but pressing in time after a breathless intense bump	an acquaintance met by chance
now a part of me, of my life, every single day, light or heartbreaking I know it will be there like the sun rising in the sky every morning	like waiting for the time to come to live and die
my companion, sometimes silent sometimes not	my pain is like a knife that pierces me
swinging in frequency a companion on the journey	my pain is like the end of the world
a fire that burns a traveling companion dull and annoying	I feel fragile like glass as if I have to be always careful of every movement, and every crack of my joints
POWERFUL AS A FIRE	like having machines running over you
a harassing master of life	the feeling that my joints are gears destined to come apart at any moment

## Data Availability

Not applicable.

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
