# Peer review of "The Language of Pain in the Hypermobile Ehlers–Danlos Syndrome: Metaphors as a Key to Understanding the Experience of Pain and as a Rehabilitation Tool"

_brainsci, 2023, doi:10.3390/brainsci13071042_

Round 1
Reviewer 1 Report
Comments and Suggestions for Authors
I think this is a timely paper on an important topic. I have no major methodological concerns, however I am not an expert in this type of analysis. I wonder whether an error was made in the abstract
'They were asked to write down what grief was to them'
Should this be pain?
Comments on the Quality of English Language
I think that the English could be improved by help from a native speaker
It is generally good but I think flow could be improved
Author Response
Than you for your comment.

Reviewer 2 Report
Comments and Suggestions for Authors
In this study, the authors the use of metaphors is a useful tool for exploring illness experience and may help clinicians in the rehabilitation program. The manuscript as a whole is good but needs revision to increase its quality.
1. In the Abstract and whole manuscript, please identify the abbreviations when firstly mentioned
2. You should carefully review your manuscript.
3. Please improve the discussion part. Comparisons are needed to justify the results.
4. Can the authors please elaborate on the significance of their findings to patients in future steps?
5. What are the limitations of the study?
6. What are your significant values?
Comments on the Quality of English LanguageEnglish structure, abbreviation and format need to be amended
Author Response
Thank you for all the comments to our paper.

Reviewer 3 Report
Comments and Suggestions for Authors
Comments to the authors
Thank you for inviting me to review the manuscript entitled “The language of pain in the hypermobile Ehlers-Danlos syndrome: metaphors as a key to understand the experience of pain and a rehabilitation tool”. This study is constituted by a cross-sectional survey sent to 89 participants diagnosed with EDS. It assesses the metaphors utilized by the participants to describe their pain, along with a correlation with anxiety and depression.
It more a qualitative report, rather than a quantitative study. However, I believe it is unique in his field, in the sense that adopts a new perspective to the experience of pain, which is normally overlooked by physicians.
Please, find here some major and minor comments that need to be addressed to improve the quality of the manuscript.
Some major concerns need to be addressed:
- the authors conclude that patients with hEDS tend to use metaphors to communicate and express their pain. However, it is not clear how many of these participants actually USED metaphors to express pain vs those who did not. Otherwise, it is not clear if the survey questionnaire was explicitly asking for a metaphor. If this was the case, than the conclusion is not supported (as the metaphor was a prompt of the survey, and not a spontaneous descriptor provided by the participant)
- I suggest that the authors reformulate the methodology. The Procedure should be moved before the Measures, otherwise the reader has not a clear idea of how the study was conducted until the end of the section. Also, the section of LIWC should be moved after the survey measures are presented, for a better interpretability of what this program is used for.
I also suggest specifying in the methods the type of study utilized. It seems to be a cross-sectional study, as it involves a survey. It is also not clear if there was a minimum duration of hEDS diagnosis. Were the participants paid to partake in the study?
Also, I find it would be very valuable and interesting if the authors could provide a further explanation of how the McGill questionnaire (that is a qualitative tool expressing the quality of pain) is transformed into a quantitative value. Could the participants select multiple choice? I suggest that the authors provide more info on this tool, especially for those readers that are not familiar with this.
It is also not clear how the initial contact occurred with the participants. Were they contacted through email? How many participants did the authors invited? Did they set up a maximum number or was it a voluntary survey data collection? Please, expand the methodology by adding all this information.
- some paragraphs currently written in the conclusion should be part of the discussion (for example those that present a comparison with the literature, like the 1st, 3rd and 4th paragraphs of page 7)
- the manuscript lacks an acknowledgment of the limitations. Some of these are for example that the diagnosis provided by the participants seemed to be self-reported, which may incur in bias or error. Other limitations may account for the voluntary participation, which should take into consideration voluntary bias (maybe overestimation or high selection of those patients with impairment and high pain intensity). From this data collection, nothing is known if these participants were currently under treatment. For example, also it would be interesting to assess the duration of the pain, especially because of the authors’ emphasize on chronic pain.
Some minor corrections are needed:
In the introduction:
The authors should clarify that EDS is a chronic pain, before presenting the definition of chronic pain. The assumption that EDS is a chronic pain is also supporting all the discussion; however, this link really needs to be clarified for a better flow and readability. Also, it would be important to add in the introduction that depression and anxiety are psychological comorbidities that often time co-present with chronic pain. This is indeed what justifies the authors to investigate anxiety and depression among the participant.
In the Results:
- all the values should have the “,” replaced with “.” For example, in the table 1, the total index is 46,65. However, different from Italian, in English and American writing, this should be 46.65 ± 13.55, and so on for all the other values.
English:
- no abbreviation should be present in the manuscript. Please, correct those
Comments on the Quality of English LanguageEnglish structure, abbreviation and format (dot instead of comma in the numbers) need to be amended throughout the text
Author Response
Thank you for all your precious comments.
